# Brain Health: Attitudes towards Technology Adoption in Older Adults

**DOI:** 10.3390/healthcare9010023

**Published:** 2020-12-28

**Authors:** Nadir G. Abdelrahman, Raza Haque, Molly E. Polverento, Andrea Wendling, Courtney M. Goetz, Bengt B. Arnetz

**Affiliations:** Division of Geriatrics, Department of Family Medicine, College of Human Medicine, Michigan State University, 788 Service Rd, East Lansing, MI 48824, USA; Raza@msu.edu (R.H.); smeltzer@msu.edu (M.E.P.); wendli14@msu.edu (A.W.); goetzcou@msu.edu (C.M.G.); arnetzbe@msu.edu (B.B.A.)

**Keywords:** innovation, technology, telemedicine, brain health, geriatrics

## Abstract

(1) *Background*: There is increasing scholarly support for the notion that properly implemented and used, technology can be of substantial benefit for older adults. Use of technology has been associated with improved self-rating of health and fewer chronic conditions. Use of technology such as handheld devices by older adults has the potential to improve engagement and promote cognitive and physical health. However, although, literature suggests some willingness by older adults to use technology, simultaneously there are reports of a more cautious attitude to its adoption. Our objective was to determine the opinions towards information technologies, with special reference to brain health, in healthy older adults either fully retired or still working in some capacity including older adult workers and retired adults living in an independent elderly living community. We were especially interested in further our understanding of factors that may play a role in technology adoption and its relevance to addressing health related issues in this population; (2) *Methods*: Two focus groups were conducted in an inner-city community. Participants were older adults with an interest in their general health and prevention of cognitive decline. They were asked to discuss their perceptions of and preferences for the use of technology. Transcripts were coded for thematic analysis; (3) *Results*: Seven common themes emerged from the focus group interviews: physical health, cognitive health, social engagement, organizing information, desire to learn new technology, advancing technology, and privacy/security; and (4) *Conclusions*: This study suggests that in order to promote the use of technology in older adults, one needs to consider wider contextual issues, not only device design per se, but the older adult’s rationale for using technology and their socio-ecological context.

## 1. Introduction

Technology has been reported to enhance and enrich the lives of older adults by facilitating better interpersonal relationships, and social connectedness positively impacting quality of life [1]. In addition, the use of technology could contribute to an improvement in physical health [2]. Age-related decline in brain health represents a great personal and financial burden to individuals, families, healthcare, and social services [3,4]. Currently, there is no known treatment for neurodegenerative diseases such as Alzheimer’s, which result in devastating consequences for the individual, family, and society. The emerging evidence points to the potential beneficial impact of lifestyle changes and cognitive training for overall health, including prevention of cognitive decline [5].

Studies link physical activity and exercise-related brain stimulation to the ability to maintain memory and learning, through an increase in hippocampal volume and improvement in spatial memory, as well as by preventing hippocampal volume loss in late adulthood, all of which contribute to retained memory function [6,7].

Newer brain health models are being developed to optimize overall general health and cognitive well-being with advanced technology [8]. Informal care provision by family members and friends is the corner stone of care in people with cognitive decline and dementia. Optimizing brain health could decrease care giving load on individual and families. Furthermore, brain health could decrease this care-giving load on already stretched healthcare systems, including skilled nursing facilities and social services [9]. The term Brain Health INnovation Diplomacy (BIND) was recently suggested by a team of diverse experts from six countries and 23 institutions. BIND, a novel working group that aims to leverage technological innovation to improve brain health, suggests it is important to bridge different determinants of health, including educational attainment, diet, access to health care, physical activity, social support, and environmental exposures in order to improve overall cognitive and physical health [10].

Based on the above recommendations from BIND, we envision that the primary approach to maximize function could be to (1) offer older persons functional and feasible tools to track their brain health and (2) implement evidence-based strategies to counteract decline. The adaptation of use of emerging electronic technology is a promising strategy to improve health outcomes and quality of life for older adults [11]. There is already a broad and increasing adoption of smart technology to track health and fitness [12,13]. The promise of using technology, e.g., apps to improve cognitive health has been subject to a few studies. However, the evidence of its impact and utility has not been proven [14].

Using technology to enhance healthcare access is a promising strategy in providing geriatric care [15] and even psychiatric evaluations and interventions [16]. Handheld personal devices allow healthcare providers to leverage technology to reach populations even more quickly and completely than ever before.

Successful adoption of technology, particularly the use of telehealth or web-based tools, depends on the end-user [17]. Older adults, in particular, have been slow to adopt technology compared to younger adults, as evidence by lower internet and broadband adoption rates in this age group [18,19]. Technology acceptance depends on the perceived usefulness and perceived ease of use of a service [20]. Knowing the importance of these two elements, understanding whether older adults would find a new technology usable is of utmost importance.

The overall objective of the current study was to determine the views and opinion towards information technologies, with special reference to brain health, in healthy older adults. We examined views regarding these issues in two groups—older adult workers and retired adults living in an independent elderly living community. We examined factors that may play a role in technology adoption and usefulness for addressing health related issues in this population.

## 2. Materials and Methods

### 2.1. Methods

We carried out a qualitative analysis of comments raised in focus group discussions regarding technology adoption and its role in detecting and addressing health concerns in older adults. Participants were recruited through brochures inviting older adults to share their views on the use of a proposed electronic tool designed to track their brain health and participants’ general views on technology. Information brochures were sent via email to a local retirement community and to working older adults identified through the Area Agency on Aging in Michigan; both groups resided in the region surrounding Lansing, MI, USA. Participation was incentivized with $20.00 gift cards to local shopping stores. The study was determined to be exempt by the Michigan State University Institutional Review Board (MSU IRB # STUDY00000554).

The research team developed focus group questions in a series of meetings focusing on the overall aims and objectives of the study. Tentative questions were tested on a small convenient sample of older persons. Based on this iterative process the team that consisted of qualitative and quantitative research experts and two faculty geriatricians developed and offered four open-ended questions (Table 1). Each focus group was led by an investigator with expertise in conducting focus groups who was accompanied by two other research team members, one of which took notes while the other one observed the group process.

The focus groups took place between August 2018 and October 2018. Participants were divided into two groups based on different background characteristics (e.g., age range, current employment status, and the area of residence). One group consisted of working adults (age range 55–62 years) who were fully employed and living independently in a metropolitan area. The other group consisted of retired adults (age range 60–80 years) living in an independent retirement community in the city of Jackson, MI, USA. In the U.S., independent living communities are housing arrangements designed exclusively for older adults, generally those aged 55 and over. At the completion of each focus group, the lead interviewer reiterated major concepts that participants had shared and asked for further comments, in order to ensure understanding and uncover any remaining themes.

Focus group interviews were recorded on audio tape and transcribed. No identifying information was collected by the research team and any identifying information disclosed by participants was removed from focus group transcriptions. The team utilized an inductive thematic analysis approach [21], with initial coding two of the co-authors, one of whom was not involved in the focus groups. The coders independently coded the focus group transcripts. The codes were combined and contrasted to develop themes thereby generating a network of associations. The themes were then reviewed and assessed for completion. Themes were individually and collectively reviewed by two co-authors and conflicts were resolved through consensus.

### 2.2. Qualitative Rigor

The consolidated criteria for reporting qualitative research (COREQ) were used to guide focus group data collection and reporting [22]. Guba and Lincoln’s criteria (creditability, transferability, dependability, confirmability) were used to achieve qualitative rigor [23]. Credibility was accomplished by using comprehensiveness during data collection and analysis. Both coders read the transcripts numerous times and thus became thoroughly familiar with the data. Transferability was ensured by presenting verbatim quotes as relevant examples given by each participant group. Dependability was assured by using one coder who was not present during the data collection. Confirmability was achieved through analyst triangulation involving multiple researchers, one of whom had not been present during the focus group discussions. All researchers analyzed the verbatim reports, then validated findings amongst themselves. See Table 1.

## 3. Results

Based on both groups combined, seven common themes emerged from the focus group interviews: physical health, cognitive health, social engagement, organizing information, desire to learn new technology, promoting technology, and privacy/security. Both groups expressed concerns about deterioration of their mental and physical health as a result of aging, and how that would affect their ability to provide for and take care of themselves. Participants in each group expressed that they currently use technology to socialize with family members and to organize and collect information, using their smartphones for things like calendar management and news and sports updates. Each focus group also expressed excitement and confidence regarding the ability to learn and use new technologies, although the older group expressed some frustration with how quickly technology continued to change. Both groups shared concerns pertinent to privacy and security using rapidly evolving technology, with information security and safety about protected health information frequently discussed among participants. Participants also shared concerns regarding the impact of technology on social relationships. The importance of ease of use and comfort with the technology in order to be effectively used was also a shared theme.

We found that working adults were more likely than retired adults to express comfort with the use of technology (e.g., fitness trackers such as Fitbit and smartwatches) and using technology daily, e.g., for doctor appointments. We also found some concerns regarding the rapid evolution of new technologies among both groups. Table 2 further explains the above-mentioned themes with examples.

## 4. Discussion

Our study explored the attitudes towards use of technology in older adults in the context of physical and cognitive health. Overall, we found positive attitudes towards the use of technology among both working and retired participants, including positivity towards current technology and its possible adaptation into their lives.

Our participants shared concerns about physical and cognitive decline with aging and were willing to explore how technology might be useful to improve and maintain health. We found that technology that improved social connections for older adults, technology that addressed perceived memory gaps such as forgetting dates, or technology that allowed for more independence such as portable EMRs [Electronic Medical Records] that allowed for travel, were already being used by participants in our working and retired age groups.

Older adults in our study also identified barriers to technology adaption and continued use. The rapid pace of technological development and challenges with adjustments were a shared concern across the groups, along with the impact of aging on the ability to use technology, which has been previously described in the literature [24]. Several of our older participants described facilitators for adoption such as younger family members who would initiate or support use. Understanding the roles of such facilitators and limiting changes to established platforms geared towards older adults were key takeaways.

It was interesting to note that themes of privacy and security were areas of concern in all participants in this study. These findings are consistent with privacy concerns raised by older adults in other studies, the most common being spam, unauthorized access to personal information, and information misuse. [25,26,27].

Most of our participants shared concerns about data safety and security in general and identified this as an impediment to the use of technology. This is an important finding, as we feel any technology designed for older adults must address this concern explicitly and should ensure and preserve the safety of all information to retain use within this population.

Our study is helpful as it identified barriers that should be addressed in the design process of any technology aimed at older adults. It also illustrated that our participants would be open to the adoption of such technology.

Limitations of this work include the small number of participants who were all from a single region, representing a well-educated college community, in the United States. As such, findings may not be generalizable to other regions or situations. Although anonymous, all focus group participants may not have felt comfortable expressing health, cognitive, or technological concerns within this group setting, and thus may have under-expressed such concerns.

In conclusion, the current focus group study involving retired and working older adults revealed a general interest in technology as it relates to cognitive engagement and brain health, as well as concerns regarding adaptation to change, data safety, and confidentiality. It is important that these complex concerns are taken into account in the design phase of technologies geared towards older adults. Understanding the needs of older adults in the context of active consumers of technology [28] will allow for more effective innovation tailored toward the needs, desires, and abilities of this growing demographic.

## Figures and Tables

**Table 1 healthcare-09-00023-t001:** Focus Group Questions.

What health problems are you most concerned about as you get older?
What are you doing or what have you done to try to prevent these health problems?
How have you used technology to prevent or manage health problems?
We are interested in helping adults maintain their brain health as they age. Would you be likely to use an app to help in maintaining your brain health? Why or why not?

**Table 2 healthcare-09-00023-t002:** Themes Identified Regarding Technology Use among Focus Groups of Older Adults.

Themes	Working Adults	Retired Adults
Physical Health	“My mother works out and is still very active, so she has been able to help my father. She has had a little issue with her hip but is still very mentally sharp.” “Start to get around our age and you figure okay that there are a couple of key things that if you fall and break an arm or something like that it can really impact your livelihood and start thinking about things like that.” “Exercise is really important. Doing it on a regular basis and making it a habit helps your focus.”	“Food is medicine. Good food is necessary to stay healthy.” “I like having my Apple Watch track my steps. The other day I was really tired and achy, but then I looked at my watch and saw I had over 11,000 steps.”
Cognitive Health	“I don’t see an end to my working, so I want to make sure that my mind is strong.” “My brother and I are at a crossroads right now of what we are seeing as dementia in my father. Is it really dementia or is it related to other health problems? It makes me worry about what is coming for me.” “Probably about three years ago, my dad forgot how to play cards, and he was always the one who kept score and now he doesn’t remember lots of things and that causes lots of anxiety.” “I want to know where I am now. I don’t want to wait until I am having problems remembering things.”	“People don’t know when they are starting to have challenges. It starts with forgetting a word or two, but then you find yourself going around the traffic circle several times because you can’t remember where you are going.” “Mental health is still very much a stigma for depression or anything like that…any type of memory loss. People don’t like to talk about it.” “A lot of older people, including me, have trouble with names. I can tell you when I meet someone in the store that I know them, but I can’t introduce them to someone else because I can’t remember their name. But I don’t know if this is because of dementia or if it is just part of getting older.” “If you begin to notice change, the first thing you think is ‘Will I lose my license? Will I be able to drive my car?’ That is really important to us.” “I think it is important that we continue to learn. I think some older people want to give up.”
Social engagement	“Our support group will fall away from us as people die and we will become more isolated.” “I think isolation leads to depression and in my parent’s age group depression leads to exhaustion.” “I am communicating and connecting with people all day long in my work. I don’t have to pretend that I enjoy having a conversation. If I am talking to someone, I am enjoying it. That’s how you learn.”	“My daughters don’t want me to call them anymore. They would rather I send an email or text.” “It’s the people who are home alone, who go out once or twice a week, that are more likely to have depression.”
Organizing Information	“My phone calendar app and the memo feature has helped me be more productive and not forget as much. I like that I can just grab my phone and make a note.” “The simplest thing we could do right now is to make our complete medical records accessible through our phones so that our doctors could see the records anywhere we need care. This is really important if we are traveling so that we can get the right care and not a lot of unnecessary tests.”	“I do all my financial work on there and I communicate with my family on the phone.” “I like the app about drug interactions. I can add in a new prescription and see if there is a problem with something I already take.” “My doctor has an app that I set up with my kids when we were all together. They have the password and can check in on what’s going on with me in case I forget to tell them something.”
Learning New Technology	“My work still keeps me sharp and I am learning new things in terms of different types of platforms. I am willing to learn a lot of new stuff in a short amount of time given where I work.” “I have tried different ways of learning. I have been doing more audio learning. I will sit and knit and have the audio book on and I can really focus this way.” “A lot of my friends are using the apps to learn new languages, so I am thinking I need to try that next.”	“I used to be really great with technology, but my son has taken over my computer and set it up the way he wants because he pays my bills. I sometimes have to call him to ask him where to find something on the computer.” “I am not good at technology, but I know how to play some games.” “The problem is if you don’t keep using it, you forget how to use it.” “I do the PowerPoint every other Sunday for church. Every once in a while, I will have two to three weeks when I don’t do it and I need to pull up the tutorial to remember how to do something.”
Rapid Advance of Technology	“In my lifetime, to go from learning how to use a slide rule to these apps I have had to learn, there is a lot of stuff that has gone through my brain.” “We are so much more connected to our kids than our parents ever were because of technology. Our parents had no idea where we were, but we can still track our kids using their phone.” “There are all these things that have changed the ways we think about things, think about what is normal in being connected to others versus being independent.”	“I have no idea how to get my computer to work. I start from zero. I don’t know anything about it. I can’t even remember my password or know how to change it.” “I used to be very comfortable with technology, but not as much now.” “I have a newer vehicle and have a device plugged in near the bottom of the dash that I can plug my phone into. When my son tells me to plug in my phone, I plug it in. This thing on the center of the dash will pop up if someone is calling me and I can push a button on my wheel to talk without having to look at my phone. When I travel, I get a lot of calls like that.”
Privacy & Security	“I am curious about those apps where you have a counselor online. I think it could be very helpful but am concerned about sharing personal information over an unsecured line.” “For people like my mom who are not as familiar with navigating healthcare…to have an app that can help them, but also need to feel comfortable with using an app to navigate healthcare without sharing information with the wrong person?”	“Don’t you attract lots of junk mail by using technology so much?” “Getting hacked can happen very easily, but my son who comes every week goes through my computer to check and see if everything is okay.” “I started doing online banking but got all sorts of warnings I was being hacked, so I stopped using it.” “My daughters have access to my computer and my iPad. I think that is important at my age. They can access everything and won’t have to search if something happens to me.”

## Data Availability

Data sharing not applicable.

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
