# Peer review of "Brain Health: Attitudes towards Technology Adoption in Older Adults"

_healthcare, 2020, doi:10.3390/healthcare9010023_

Round 1

Reviewer 1 Report

Overall:
The topic of this manuscript is important, but I do not feel that it goes into sufficient detail. This ms seems more like part 1 of a multi-method study. As a stand-alone manuscript, it doesn't seem likes there’s enough here to warrant a journal article.

Here’s one example where the data collected doesn’t support claims in the discussion:

“Our study is helpful as it identified barriers that should be addressed in the design process of any application technology for brain health tracking and improvement.”

This claim is not supported. The technology adoption research community already knows that there are barriers to technology adoption among older people. I did not read anything novel in the Results section in support of the claim.

The topic is definitely important. I recommend that additional, more targeted research be conducted to flesh out and deepen the contributors’ discoveries in this area, which would allow for a more expansive writeup by the authors.

Line items:
10: “Studies show cognitive and physical exercise could be helpful innovative approaches towards brain health.” - Grammar - not sure of meaning, but something belongs between “helpful” and “innovative”.

11: “Use of technology has potential to improve engagement and promote cognitive and physical health.” - Recommend “Use of technology such as [broad name or category] has potential…”

14: “Our objective was to determine the views and opinion…” - Grammar - should be “opinions”.

25: Double period.

42-47: Run-on sentence. Shorten or split. It also seems like the last few dependent clauses are not constructed in parallel.

44: Is “diplomacy” a field-specific word in this context, or does it simply mean “intervention”?

57: “Handheld personal devices allow a synergistic route for healthcare and technology to combine…” - Phrasing. Suggest “Handheld personal devices allow healthcare to leverage technology…” or something similar.

Author Response

Authors’s Responses

Reviewer

Author response

Overall:

The topic of this manuscript is important, but I do not feel that it goes into sufficient detail. This ms seems more like part 1 of a

multi-method study. As a stand-alone manuscript, it doesn't seem like there’s enough here to warrant a journal article.

Here’s one example where the data collected doesn’t support claims in the discussion: “Our study is helpful as it identified barriers that should be addressed in the design process of any application technology for brain health tracking and improvement.” This claim is not supported.

The technology adoption research community already knows that there are barriers to technology adoption among older people.

I did not read anything novel in the Results section in support of the claim.

We have added information to different sections to address the concerns of the reviewers and ensure the study was designed to stand alone.

We rewrote the discussion to avoid any unsubstantiated claims. New references were also added to support this work and help delineate its niche in the existing literature.

The topic is definitely important. I recommend that additional, more targeted research be conducted to flesh out and deepen the contributors’ discoveries in this area, which would allow for a more expansive writeup by the authors.

Thank you, we have rewritten the introduction and discussion to better frame this work and added further details about the identified themes to inform the reader.

10: “Studies show cognitive and physical exercise could be helpful innovative approaches towards brain health.” - Grammar - not sure of meaning, but something belongs between “helpful” and “innovative”.

Sentence was rewritten

11: “Use of technology has potential to improve engagement and promote cognitive and physical health.” - Recommend “Use of technology such as [broad name or category] has potential…”

Addressed

14: “Our objective was to determine the views and opinion…” - Grammar - should be “opinions”.

Addressed

25: Double period.

Fixed. Thank you.

42-47: Run-on sentence. Shorten or split. It also seems like the last few dependent clauses are not constructed in parallel.

Fixed. Thank you

44: Is “diplomacy” a field-specific word in text, or does it simply mean “intervention”?

This was changed to working group to more accurately describe the effort

57: “Handheld personal devices allow a synergistic route for healthcare and technology to combine…” - Phrasing. Suggest “Handheld personal devices allow healthcare to leverage technology…” or something similar.

Addressed. Thank you.

Reviewer 2 Report

This qualitative paper outlines the perspective of healthy older people towards information technologies, namely regarding to brain health. Thus, this work adds to the body of knowledge, the important perspective of older people about the use technologies in their lives.

Broad comments

This is an important and timely topic. Plus, giving voice to older people on topics about them, is of great importance. However, this manuscript needs to be improved, specifically it needs to provide more information on methodology (with references) and, on discussion section, data should be interpreted in light of what is already known on the topic.

Specific comments

Introduction

Line 29: please consider change “suffering from declining cognition” to a more positive language, e.g., “experiencing cognitive declines”.

Lines 41-42: Informal care provision is the cornerstone of care in people with cognitive decline in must world regions. Optimizing brain health could also decrease the load and burden on individuals and families?

Line 61: The authors state that older people have been slow to adopt technology. How much slow? Are they being slow to adopt technology or is it difficult to follow the rapidly development of technology?

Lines 67-68: “Retired adults living in an independent elderly living community” were included in this study. Please clarify what is an independent elderly living community, perhaps on methods section.

Materials and methods

Line 85: Please provide examples of associated prompts.

Line 94: Were the interviews recorded just on audio? Or video too? Please clarify.

Lines 96-99: Did the authors use a software for coding? If yes, provide more information about it please.
It seems like an inductive thematic analysis was used. Please clarify and provide references for the methodological approach used.

Results

Page 3: “(Fitbit, smart phones, etc)”. With “Fitbit” do you mean smartwatches? Or other gadgets too?

Table 2: please format table 2.

Discussion

Lines 140-146: Please discuss this results, and compare them with the most recent available literature on the topic.

Lines 147-148: Were these barriers identified before (i.e., other studies, with similar population)? How can these barriers be addressed?

Line 151: Have these concerns been highlighted in the literature?

Lines 152-153: How was the sample size defined? Did the interviews achieve data saturation? This information should be clarified on methods section.

Limitations

Line 161: The “overlooked in many prior technology users studies” should be used to discuss the results of this study and references should be provided.

Author Response

Reviewer comment

Author response

Reviewer 2

Comments and Suggestions for Authors: This qualitative paper outlines the perspective of healthy older people towards information technologies, namely regarding brain health. Thus, this work adds to the body of knowledge, the important perspective of older people about the use of technologies in their lives.

Thank you.

Broad comments: This is an important and timely topic. Plus, giving voice to older people on topics about them, is of great importance. However, this manuscript needs to be improved, specifically it needs to provide more information on methodology (with references) and, in the discussion section, data should be interpreted in light of what is already known on the topic.

Thank you. We have provided additional and more detailed information in the methods section. The discussion section has been extensively rewritten to address reviewer critique.

Line 29: please consider changing “suffering from declining cognition” to a more positive language, e.g., “experiencing cognitive declines”.

Addressed - changed.

Lines 41-42: Informal care provision is the cornerstone of care in people with cognitive decline in must world regions. Optimizing brain health could also decrease the load and burden on individuals and families?

Addressed. Thank you.

Line 61: The authors state that older people have been slow to adopt technology. How much slower? Are they being slow to adopt technology or is it difficult to follow the rapid development of technology?

More information was added to address and clarify this concern, and an additional study was cited in support.

Lines 67-68: “Retired adults living in an independent elderly living community” were included in this study. Please clarify what is an independent elderly living community, perhaps in the methods section.

Clarification added. Thank you.

Materials and methods

Line 85: Please provide examples of associated prompts.

This portion of the statement was removed as no additional prompts were used.

Line 94: Were the interviews recorded just on audio? Or video too? Please clarify.

Audio only. Clarification added.

Lines 96-99: Did the authors use a software for coding? If yes, provide more information about it please.

It seems like an inductive thematic analysis was used. Please clarify and provide references for the methodological approach used.

No software was used for coding.

Yes, inductive thematic analysis was used. Clarification and references added.

Results: Page 3: “(Fitbit, smart phones, etc)”. With “Fitbit” do you mean smartwatches? Or other gadgets too?

Addressed. Clarification added.

Table 2: please format table 2.

Done.Thank you

Discussion: Lines 140-146: Please discuss this results, and compare them with the most recent available literature on the topic.

We reframed our discussion to more closely reflect our findings and relate findings to the with existing literature.

Lines 147-148: Were these barriers identified before (i.e., other studies, with similar population)? How can these barriers be addressed?

We added information from previous studies that confirmed that some of these barriers had been identified before, although many were novel. We also discussed possible means to address these barriers, although a more definite solution would require more detailed studies.

Line 151: Have these concerns been highlighted in the literature?

Our findings supported similar findings from other studies. We added this information.

Lines 152-153: How was the sample size defined? Did the interviews achieve data saturation? This information should be clarified in the methods section.

As these were pre-determined focus groups, we did not continue interviews. We did clarify themes with participants prior to completion of each group and encouraged additional comments to ensure themes of concern to participants were identified.  This information was added to the manuscript methods.

Limitations: “technology users studies” should be used to discuss the results of this study and references should be provided.

Thank you for this important point. This was added to the conclusion sentence and was referenced accordingly.

Round 2

Reviewer 2 Report

Thank you for your effort to improve this work.
I have no more comments or suggestions to add.